# Design and Mechanical Characterisation of a Large Truss Structure for Continuous Manufacturing in Space

**DOI:** 10.3390/ma15176025

**Published:** 2022-09-01

**Authors:** Peng Li, Hongyang Ning, Jiayong Yan, Bo Xu, Hongjian Li

**Affiliations:** 1Mechanical Engineering College, Yanshan University, Qinhuangdao 066004, China; 2Hebei Innovation Center for Equipment Lightweight Design and Manufacturing, Qinhuangdao 066004, China; 3Mechanical Engineering College, Beihua University, Jilin 132022, China; 4Beijing Spacecraft Manufacturing Co., Ltd., Beijing 100094, China

**Keywords:** in-space manufacturing, truss braiding, mechanical properties

## Abstract

In this paper, large space structures are essential components of significant equipment in orbits, such as megawatt-class solar power plants and long baseline interferometry. However, to realize the in-space fabrication of such megastructures, the primary consideration is the continuous fabrication of the structure. In this paper, we propose and design a structural form that differs from the minimum constituent unit shape of conventional truss structures by using an efficient winding and weaving method to construct truss structures. The continuously buildable one-dimensional truss’s structural design and mechanical properties are investigated. The parameters affecting the fundamental frequency of the truss structure are analyzed through modeling, simulation and experimental verification of the continuously buildable 1D truss. It is concluded that this configuration truss can be built continuously in space. The most influential factors on the fundamental truss frequency are the truss section spacing, the total truss length and the truss-specific stiffness. The simulated and theoretical values of the truss’s static stiffness and vibration frequency have minor errors, which provide a basis for the configuration design for the continuous manufacturing of large truss structures in space.

## 1. Introduction

Large truss structures have potential applications in space telescopes, long-baseline space observers, space solar power plants, etc. With the advancement of significant truss structure technology, the process of space exploration can be significantly advanced [1,2,3]. The primary method of obtaining large truss structures is still predominantly by folding and spreading, i.e., ground assembly for collection and launch and in-space deployment for service [4,5,6]. The fold-and-spread process is somewhat limited as the truss size increases due to the space available to carry the rocket, the high cost of construction and low space utilization. In-space fabrication [7,8,9,10,11] is an inevitable trend in the construction of large space truss structures, i.e., the material is packaged for launch on the ground and fabricated for connection in space so that the continuous fabrication of truss structures is one of the necessary conditions for the construction of substantial structural dimensions.

Constructing large trusses in space using component assembly can effectively improve space utilization, reduce launch costs and break the limit on the total truss length. The main methods of forming trusses from raw materials at this stage are winding and component joint starting. The Trusselator [12,13,14] prototype developed by Tethers Unlimited Inc. makes rods by extrusion and then connects them to become a truss, mainly considering the problem of truss stiffness. The Beambuilder truss [15] fabrication machine researched by General Dynamics consists of stringers, vertical rods and diagonal rods connected to form trigonometric truss trusses, which are not easily fabricated continuously due to the presence of vertical rods. TUI uses 3D printing technology [16] to print parts of trusses for splicing, but the manufacturing efficiency is low. Tae-Hyum Kim et al. [17] proposed an foldable truss structure in the form of a flat plate for storage, effectively increasing the packing density. Still, its unfolding structure was serialized by multiple hinges and had complex movements. Lu Dai et al. [18] proposed a new expandable truss structure for large aperture antennas, which was analyzed and topologically optimized. Finally, they concluded that the smaller the thickness and the larger the diameter of the cube cross-section, the greater the structure’s stiffness. Jong-Eun Suh et al. [19] proposed a two-dimensional modular deployable truss structure by comparing the dynamical model with the experimental model; it was concluded that it could accurately describe the behavior of the proposed deployable system. Li et al. [20] established finite element equations for a sizeable toroidal truss structure under the action of temperature and optical pressure fields. The analysis found that the effect of optical pressure on deformation was minimal. Finite element analysis of a large space truss by B. Siriguleng et al. [21] showed that the energy within the truss would be transferred from low-order to high-order modal vibrations when the excitation frequency coincided with the low-order frequency.

Although truss configuration design has been studied in most references, the configuration design of large trigonometric trusses prepared by winding and weaving is less studied. This study uses an efficient winding and weaving method to build large trusses and uses fewer material types. This can effectively improve space utilization, reduce launch costs, break the limit of total truss length, and enable continuous manufacturing in space. The vibration characteristics of the truss are one of the important conditions for the serviceability in the rail. The paper aims at the configuration design of one-dimensional truss structure that can be constructed continuously. Section 2 introduces the mechanical modelling of the truss, Section 3 introduces the simulation analysis of the truss, and Section 4 introduces the experimental verification of the truss’s static stiffness and vibration frequency. The specific stiffness, section spacing and overall length of the joist influence the vibration frequency of the post.

## 2. About Large Structures in Space Manufacturing Methods

### 2.1. Structural Design Constraints

Design constraints for the in-space fabrication of large truss structures include the following four areas.

(1)Ambient temperature suitability of raw materials, which are subject to large differences in temperature under track conditions, with a temperature variation range of ±150 °C. The selected raw materials should be adapted to the changes in space temperature.(2)Packing density of upstream replenishment material. Upstream replenishment materials should be consistent in configuration and size as much as possible to maximize the utilization of the volume of the material in the box, that is, to obtain a greater packing density.(3)Types of materials that make up the structure. The less variety of materials required for the truss structure, the fewer external resources are required for in-space fabrication and the simpler the construction process.(4)Applicability of the joining process between the elements. The in-space connection between elements should be low power consumption, low force, low thermal disturbance and low dependence on the space environment. No additional connectors are needed to minimize the consumption of space resources in the in-space connection process.

### 2.2. Raw Material Selection

The materials commonly used for the ground erection of continuously buildable one-dimensional truss structures are structural steel (45#), aluminum alloy (7075), polyether ether ketone (PEEK) and polycarbonate/carbon fiber (PC/CF), which vary greatly by temperature under track conditions, with temperature variations between ±150 °C. The glass transition temperature or melting point, thermal conductivity, low-temperature environmental suitability and manufacturing process of the four materials are compared to select the optimal material, as shown in Table 1.

As can be seen from Table 1, PEEK is the most suitable material for in-space fabrication, and the glass transition temperature can be further increased to above 150 °C by increasing its crystallinity.

### 2.3. In-Space Connection Method

Large truss structures used for in-space service are usually non-disassembled structures, so their connection processes are usually chosen as non-disassembled. Considering the constraints on the applicability of in-space joining processes between materials, ultrasonic welding and induction welding have a high potential for in-space joining applications due to their low power consumption, low force, low thermal disturbance and low dependence on the space environment, and no need for additional connectors.

Ultrasonic welding, as shown in Figure 1, under the conditions of applied pressure, use high-frequency vibration of the welding machine joints to transfer a high-frequency vibration wave to the surface of two or more welded objects so that the high-frequency friction between the surface of the object to be welded, the surface of the molecular layer melting and then solidification, form a solid connection. It has the advantage of low power and a small heat-affected area. This paper selects ultrasonic welding, and the main technical parameters are: vibration frequency 28 kHz, power 700 W, welding time 0.5 s, holding time 0.5 s, welding pressure 40 N, and vibration amplitude 30 μm. Induction welding, shown in Figure 1, involves placing metal inserts between the bonded plastic surfaces, temporarily bonding them with appropriate pressure, and placing them in a high-frequency magnetic field. The metal insert melts the plastic due to induction heating, and the plastic parts are joined by cooling. It is characterized by a simple structure and high reliability.

### 2.4. In-Space Construction Program for Large Structures

Taking into account the truss structure constraints, a scheme of in-space braiding of the truss structure based on strip material is designed, and its schematic diagram is shown in Figure 2. The truss on rail construction system mainly consists of a raw material storage module, chord rod supply module, tilt rod winding module, truss inner support module and node connection module. Left tilt rod material 7 is pulled out from left tilt rod material storage tray 6; left tilt rod material storage tray 6 is fixed on left tilt rod rotating frame 5 and rotates left with left tilt rod rotating frame 5. During the rotation, left tilt rod material 7 will be wound on chord rod 12. In the same way, suitable tilt rod material 10 will be wrapped around stringers 12. While feeding the whole truss 15 forward, the connecting and fixing device 11 will connect and fix the interface area of chord 12, left-tilt rod 13 and right-tilt rod 14, and then output fixed truss 15 outward.

Where 1 is the body; 2 is the string raw material storage tray; 3 is the string raw material; 4 is the string feeding wheel; 5 is the left tilt rod rotating frame; 6 is the left tilt rod raw material storage tray; 7 is the left tilt rod raw material; 8 is the right tilt rod rotating frame; 9 is the right tilt rod raw material storage tray; 10 is the right tilt rod raw material; 11 is the connecting cementing device; 12 is the string; 13 is the left tilt rod; 14 is the right tilt rod; 15 is the truss. 

This solution requires only one material for upstream replenishment (strip material): high packing density (strip material is packed in the form of material rolls, simple configuration, and high space utilization); ultrasonic welding is selected for the connection process, and the in-rail connection process is good; PEEK is selected as the raw material, and the in-rail selection is applicable. In the ground test phase, because PEEK is expensive, according to the parameters in Table 1, PC/CF can also meet the requirements of ground tests and has a certain degree of stiffness. PC/CF is selected instead of PEEK to perform the ground demonstration test in this paper.

## 3. Design and Mechanical Modelling of Continuously Buildable 1D Truss Configurations

### 3.1. Analysis of the Necessary Conditions for the Configuration of a One-Dimensional Truss That Can Be Built Continuously

The truss can be divided into vertical and non-vertical types, usually consisting of webs and chords, with the central systems of traps being herringbone, cross, K-shape and diamond. Compared to the component connection, the winding method is relatively simple and only requires the twisted head to be controlled to follow a predetermined trajectory. Therefore, this paper adopts the spiral approach for constructing herringbone trusses without vertical rods.

As shown in Figure 3, to ensure the continuity of the winding process, it should be provided that termination point D of this loop and start point E of the next loop coincide after one continuous winding of the fibers. As shown in Figure 4, triangle ABC is a repeating unit of the truss, rods AB and AC are the webs of the truss and rod BC is the chord of the truss. Let the pitch of the truss be *l*. The horizontal spacing A between the upper vertex A and the left vertex B of triangle ABC is called the left span $l_{l}$. Similarly, the right span is $l_{r}$. The ratio of the left span to the pitch of the truss becomes the left span ratio, noted as x, $x=l_{l}/l$. According to Figure 3, the left span is two times the proper span, i.e., x=2/3; similarly, the right span can be two times the left span, i.e., x=1/3.

The continuously windable herringbone web system of trusses constructed in this paper is 2 left-handed diagonal rods and 1 right-handed diagonal rod. The truss structure is shown in Figure 5, where the red rods are right-handed tilt rods, and the green and blue rods are left-handed tilt rods.

### 3.2. Modelling of Mechanical Properties of Continuously Buildable Truss Structures

(1)Analysis of the mechanical properties of the truss

For large extra-long trusses, the structure has a relatively large length and diameter and approximates a flexible rod structure. The fundamental frequency analysis can be reached as a continuous beam. The first order angular frequency equation [12,24] for a constant beam structure with free ends is:(1)$$\omega_{1}=\frac{4.730^{2}}{L^{2}}\sqrt{\frac{EI}{m}}$$
where *L* is the total length of the beam (m); *E* is the Young’s modulus of the material (Pa); *I* is the cross-sectional moment of inertia of the beam (m^4^); *m* is the linear density of the beam (kg/m).

First-order vibration frequency:(2)$$f_{1}=\frac{\omega_{1}}{2\pi }=\frac{4.730^{2}}{2\pi L^{2}}\sqrt{\frac{EI}{m}}$$

Wire density:(3)$$m=m_{l}+m_{ld}+m_{rd}$$
where $m_{l}$ is the chord line density (kg/m), $m_{ld}$ is the lefthand slant line density (kg/m) and $m_{\mathrm{rd}}$ is the right-hand slant line density (kg/m).
(4)$$m_{l}=3\times \frac{\pi d_{l}{}^{2}}{4}\times \rho =\frac{3\pi d_{l}{}^{2}\rho }{4}$$
(5)$$m_{ld}=3\times \frac{\pi d_{ld}{}^{2}}{4}\times \rho \times \frac{\sqrt{{\left(xl\right)}^{2}+b^{2}}}{xl}=\frac{3\pi d_{ld}{}^{2}\rho \sqrt{{\left(xl\right)}^{2}+b^{2}}}{4xl}$$
(6)$$m_{rd}=3\times \frac{\pi d_{rd}{}^{2}}{4}\times \rho \times \frac{\sqrt{{\left(1-x\right)}^{2}l^{2}+b^{2}}}{(1-x)l}=\frac{3\pi d_{rd}{}^{2}\rho \sqrt{{\left(1-x\right)}^{2}l^{2}+b^{2}}}{4(1-x)l}$$
where *b* is the truss section side length, *l* is the truss section pitch, $d_{l}$ is the truss chord diameter, $d_{ld}$ is the left-hand tilt rod diameter and $d_{rd}$ is the right-hand tilt rod diameter.

When $d_{l}=d_{ld}=d_{rd}=d,x=\frac{2}{3}$,
(7)$$m=m_{l}+m_{ld}+m_{rd}=\frac{l+\frac{3}{2}\sqrt{{\left(\frac{2l}{3}\right)}^{2}+b^{2}}+3\sqrt{{\left(\frac{l}{3}\right)}^{2}+b^{2}}}{l}m_{l}$$

(2)Moment of inertia of the truss section

Neglecting the effect of the moment of inertia on the joist section generated by the webs, only the moment of inertia of the section generated by each chord section is calculated, and the cross-section of the joist is shown in Figure 6.
(8)$$b=\sqrt{3}D/2$$
(9)$$A=3A_{l}$$
(10)$$\varphi_{i}=i\cdot 2\pi /3+\alpha \;\left(i=0,1,2\right)$$
(11)$$I_{y}=\int_{A}z^{2}dA=\sum_{i=0}^{2}z_{i}^{2}\cdot \frac{A}{3}={\frac{D}{4}}^{2}\cdot \frac{A}{3}\sum_{i=0}^{2}\sin^{2}\varphi_{i}$$

Substituting Equations (8)–(10) into (11) gives
(12)$$I_{y}=A\cdot D^{2}/8$$
(13)$$I=I_{y}=I_{z}=A\cdot b^{2}/6$$
where *A* is the total cross-sectional area of the chord; $\varphi_{i}$ is the angle between the center of the ith chord section and the center of the truss section and the Y-axis; D is the diameter of the outer circle of the truss section; $z_{i}$ is the distance between the center of the ith chord and the Z-axis; α is the angle between the center of the first chord section and the center of the truss section and the Y-axis; $I_{y}$ and $I_{z}$ is the moment of inertia of the chord section about the Y-axis and the Z-axis.

(3)Truss vibration frequency

Substituting Equations (7) and (13) into Equation (2) gives
(14)$$f_{1}=\frac{4.730^{2}b}{2\pi L^{2}}\sqrt{\frac{E}{6\rho \left[1+\frac{3}{2}\sqrt{\frac{4}{9}+{\left(\frac{b}{l}\right)}^{2}}+3\sqrt{\frac{1}{9}+{\left(\frac{b}{l}\right)}^{2}}\right]}}$$

To ensure that the number of repeating units N of the joist is an integer, the joist section spacing *l* is 50 mm, 75 mm, 100 mm, 125 mm, 136.4 mm, 150 mm, 166.7 mm, 187.5 mm, 214.3 mm, 250 mm and 300 mm, respectively. The joist material is chosen from structural steel, aluminum alloy, PEEK and PC/CF, respectively. Figure 7 shows the change curve of fundamental frequency for different materials. With the constant total length of the joist, section edge length and diameter of the joist rod, the first order vibration frequency of the beam increases as the pitch of the joist increases. The fundamental frequency change of different materials is also other, PEEK vibration frequency is the least and CF/PC vibration frequency is the most.

(4)Static stiffness

Bending stiffness of cantilever beams:(15)$$EI=\frac{FL^{3}}{3w}=\frac{mgL^{3}}{3w}$$
(16)$$\frac{w}{m}=\frac{1}{2}(k+k^{’})$$

Flexural stiffness:(17)$$EI=\frac{\pi b^{2}d_{l}{}^{2}}{8}E$$
where *m* is the mass of the loaded weight; *g* is the acceleration of gravity, taken as 9.8 m/s^2^; *L* is the cantilever length; *w* is the end deflection; *k*, *k*’ is the slope of the deflection displacement curve; *b* is the side length of the truss section; *d_l_* is the diameter of the truss chord.

## 4. Simulation of the Mechanical Properties of a Continuously Buildable Truss Structure

Abaqus finite element analysis software is used for modal analysis. The truss model was a line body model, the total length *L* of the truss was tentatively set to 1500 mm, the section spacing *l* to 50 mm and the section edge length to 75 mm. In the assembly process, the completed line body model is combined into a whole; the analysis step is set to the frequency in linear regression, and the first 10 frequencies are taken to facilitate the observation of the vibration characteristics of the first 4 orders. When the mesh is divided, it is chosen to be laid on edge with the number of 2, and the cell type is chosen to be a beam cell; finally, the job is created and submitted. When doing static stiffness simulation, the basic steps are the same as modal analysis, and the analysis step is selected as the static general analysis step; the boundary condition is set to be fixed entirely at one end, and the load is applied at the other end, and the load type is concentrated force. The relationship between the joist vibration frequency f and the structural parameters of the joist is investigated with the joist section spacing l, the joist rod diameter d and the total length L as variables.

### 4.1. Material Influence on Truss Fundamental Frequency

(1)Poisson’s ratio

Different values of Poisson’s ratio μ are chosen for structural steel materials, and simulations are carried out on truss structures to investigate the effect of material Poisson’s ratio μ on the fundamental frequency of the truss. The simulation results are shown in Figure 8. The Poisson’s ratio is 0.2, 0.3 and 0.4. From the figure, it can be concluded that the variation of fundamental frequency under different Poisson’s ratio μ is minimal, the variation is less than 0.30% and the effect of material Poisson’s ratio on the fundamental frequency of joist vibration can be ignored.

(2)Specific stiffness

Modal analysis was carried out on structural steel, aluminum alloy, PC/CF and PEEK materials to investigate the effect of material-specific stiffness $\sqrt{\frac{E_{i}}{\rho_{i}}}/\sqrt{\frac{E_{s}}{\rho_{s}}}$ on the fundamental frequency of the truss. The simulation results are shown in Figure 9. The fundamental frequency of PC/CF is the largest, and that of PEEK is the smallest.

### 4.2. Influence of Truss Configuration Parameters on Mechanical Properties

(1)Radius of truss rod section

The joist pitch l is 75 mm, the cross-sectional diameters of the joist rods are 2 mm, 2.5 mm, 3 mm, 3.5 mm and 4mm, respectively, and the joist material is chosen as PC/CF. The simulation results are shown in Figure 10. In the selected parameter range, the first-order bending frequency of the joist increases slowly with the increase of the section diameter. Still, the variation range is 0–0.3 Hz, which is negligible. Therefore, the truss rod section diameter does not affect the vibration frequency change of the truss.

(2)Section spacing

The overall length of the joist is 1500 mm, the pitch of the beam is 50 mm, 75 mm, 10 mm, 125 mm, 136.4 mm and 150 mm, and the beam’s material is structural steel, aluminum alloy, PC/CF and PEEK. The simulation results are shown in Figure 11. The first order is bending frequency of the joist increases as the pitch of the joist increases.

(3)Total length of truss

The joist section spacing l is 75 mm, the total length of the joist is 1500 mm, 2025 mm, 2250 mm, 2475 mm, 3000 mm, 4500 mm and 6000 mm, respectively and the material of the post is PC/CF. The simulation results are shown in Figure 12. The first-order bending frequency of the beam decreases as the length of the joist increases.

## 5. Experimental Validation and Analysis

In this paper, the truss structure is constructed by winding and weaving, and the herringbone truss structure without a vertical rod is prepared. The truss samples’ static stiffness and vibration characteristics are tested separately, the truss’s bending stiffness and vibration characteristics are verified, and the reasons affecting the stiffness and vibration frequency of the truss are analyzed.

### 5.1. Truss Stiffness Test

The experimental setup for the static stiffness test of the truss is shown in Figure 13. The truss construction system prepared the 2 m long continuous truss sample. The truss is fixed with weights at 0.8 m of the truss sample, and weights are hung at the free end of the truss sample with a starting weight of 40 g and increased to 320 g with a cut-off weight of 40 g each time. The deflection-deflection curve at the free end of the truss is measured by using displacement sensors. The deflection-deflection curve of the truss sample is shown in Figure 14.

The slope of the curve can be derived from Figure 14, and the slope and L = 0.8 m are substituted into Equations (15) and (16) to obtain $\mathrm{EI}=613.3\;N\cdot m^{2}$. Substituting b = 75 mm, dl = 2 mm and $E_{CF/PC}=7.0\times 10^{10}\;\mathrm{Pa}$ into Equation (17) to get the theoretical value of joist flexural stiffness $EI_{t}=618.19\;N\cdot m^{2}$. With the increasing mass of the loaded weight, the deflection value of the joist sample also increases. During this change, the connection defects at the nodes of the joist sample have a more significant influence on the stiffness of the joist.

The theoretical, tested and simulated values of flexural stiffness of the truss sample are shown in Figure 15. The error between the tested and theoretical values of the truss is 5%, and the simulated value is lower than the tested value. Possible reasons for the errors: the actual modulus of elasticity is smaller than the reference value because the modulus of elasticity of the material is not directly measured; defects in the prepared joist samples, such as breakage of some fibers or insufficient straightness of the rod; and defects in the nodes of the joist samples, such as some nodes are not effectively connected.

### 5.2. Truss Vibration Characteristics Test

The experimental setup for truss vibration characteristics testing is shown in Figure 16. A 2 m-long continuous truss sample is prepared using the truss construction system, and the truss sample is placed on the vibration platform and fixed with a heavyweight. The vibration sensor is specified on the truss, and the vibration frequency of the vibration platform is set to 0–300 Hz to sweep the truss sample. Finally, the truss’s first four-order frequency values were obtained, and the results are shown in Table 2.

Considering the influence of the sensor wiring, the point mass load of the sensor was set to 20 g in the simulation, and the results of the comparison between the simulated and tested values are shown in Table 2. In the modal simulation of the joist, the first-order to fourth-order bending frequencies of the beam is 76.44 Hz, 76.45 Hz, 203.41 Hz and 203.49 Hz, respectively, which are closer to the test values, but all of them are higher than the test values. The error between the test value of joist vibration frequency and the simulated value may be caused by the actual elastic modulus being smaller than the reference value because the elastic modulus of the material is not measured directly; there are defects in the prepared joist sample, such as fracture of some fibers or insufficient straightness of the rod; and there are flaws in the nodes of the joist sample, such as some nodes are not connected effectively.

## 6. Conclusions

In this paper, after sorting out the basic principles of in-space manufacturing, the truss configuration design and truss fundamental frequency analysis are carried out, the actual frequency formula of the trigonometric truss is established the truss construction system is designed by using the winding and braiding method and the modal analysis and static stiffness analysis are carried out by finite element analysis. The test platform is built to test the truss samples. Specific conclusions were drawn as follows.

(1)The truss construction system can be used to prepare continuous truss samples so that the carbon fiber-reinforced auxiliary materials can be manufactured continuously without interruptions to improve their stiffness and the truss construction efficiency.(2)The fundamental frequency of the truss is independent of Poisson’s ratio of truss material and section diameter d. The fundamental frequency of the joist is related to the overall length L, section spacing l and material-specific stiffness of the beam.(3)The measured flexural stiffness is 5% smaller than the theoretical value by the static stiffness test and joist vibration fundamental frequency test. The error between the measured and simulated values of the first two vibration frequencies is 5%.(4)The error may be caused by the fact that the elastic modulus of the material is not directly measured, resulting in the actual elastic modulus being smaller than the reference value; defects in the prepared truss samples, such as the breakage of some fibers or insufficient straightness of the rod frame; defects in the nodes of the truss samples, such as some nodes failing to connect effectively.

Future research work to be carried out: (1) in-depth study of the mechanical properties of large space truss structures in orbital service and carry out experimental verification; (2) the establishment of a truss node ultrasonic welding quality model so that the simulated and experimental values of the mechanical properties of the truss are closer.

## Figures and Tables

**Figure 1 materials-15-06025-f001:**
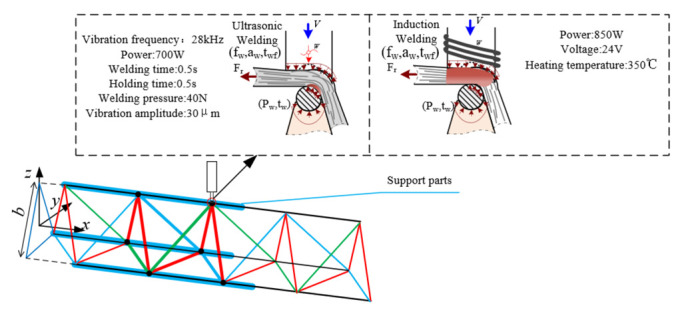
Welding method diagram.

**Figure 2 materials-15-06025-f002:**
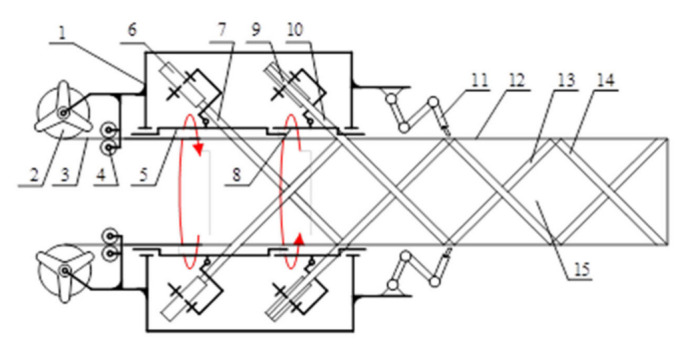
Structural composition of the truss construction prototype.

**Figure 3 materials-15-06025-f003:**
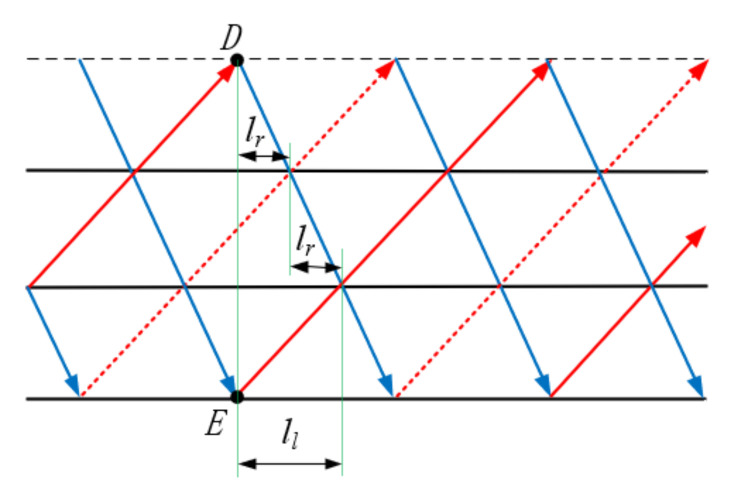
Truss expansion diagram.

**Figure 4 materials-15-06025-f004:**
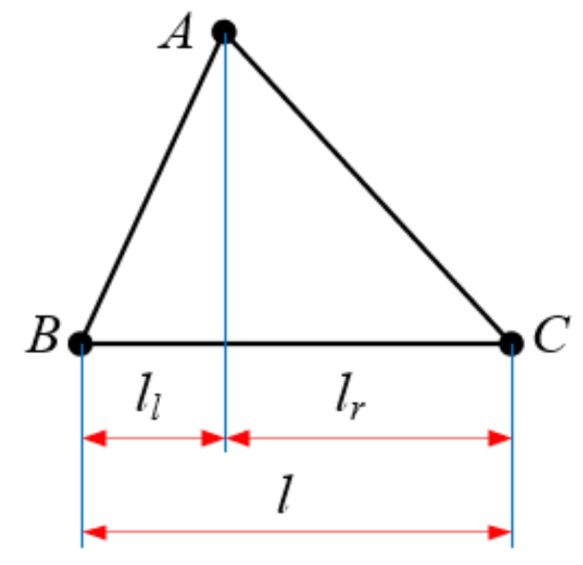
Repeating unit of truss.

**Figure 5 materials-15-06025-f005:**
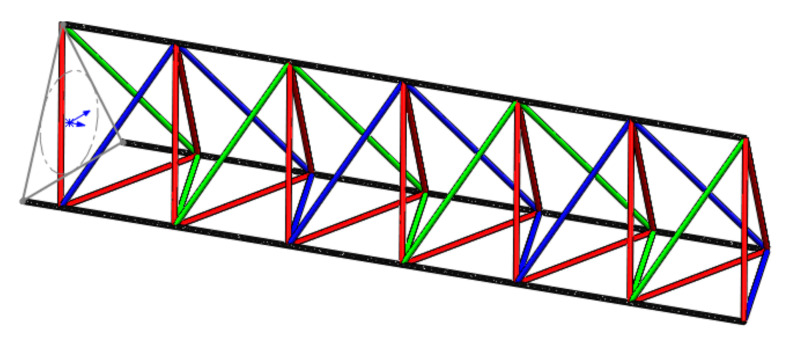
Trigonometric truss structure.

**Figure 6 materials-15-06025-f006:**
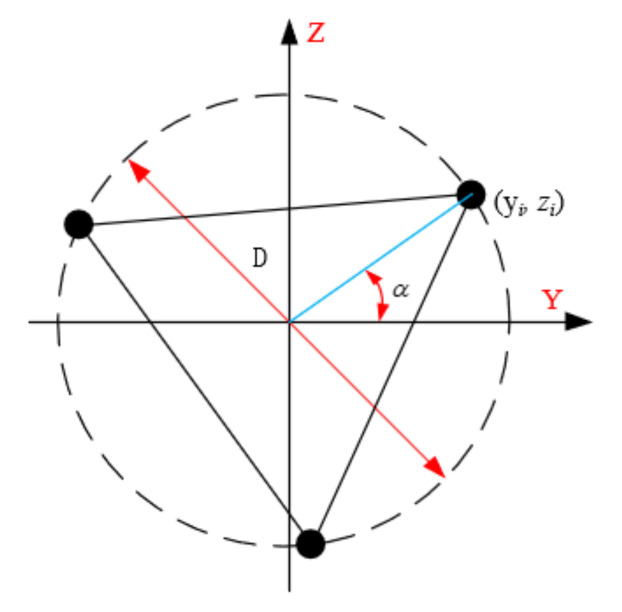
Section of the truss.

**Figure 7 materials-15-06025-f007:**
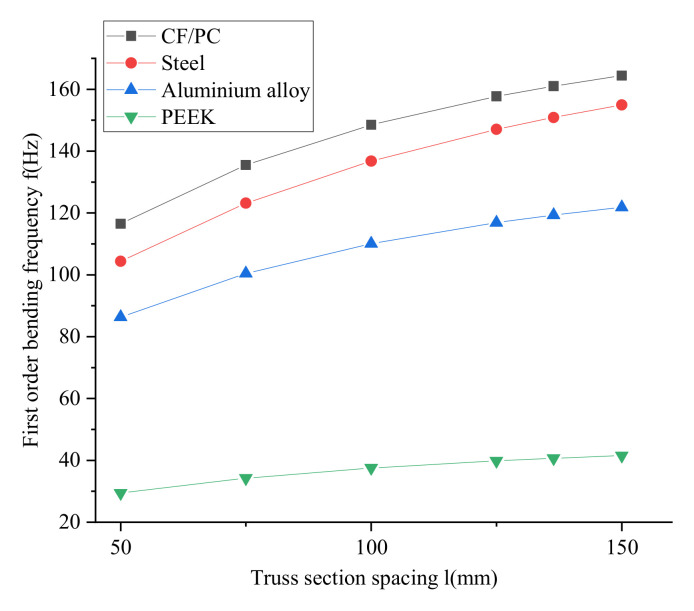
Fundamental frequency variation curves for different materials.

**Figure 8 materials-15-06025-f008:**
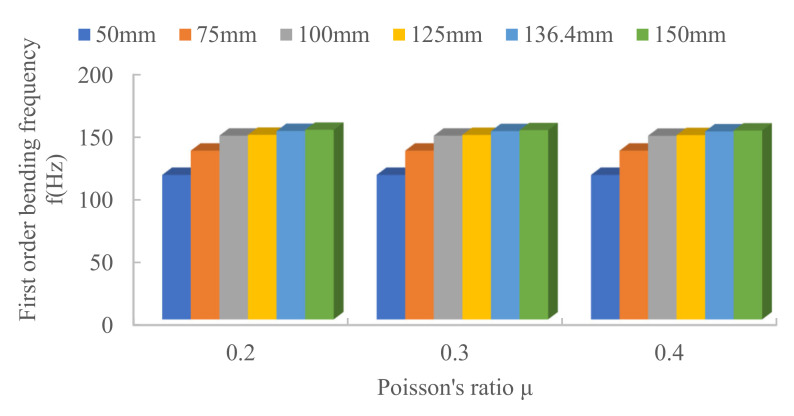
Effect of Poisson’s ratio on the fundamental frequency of the truss.

**Figure 9 materials-15-06025-f009:**
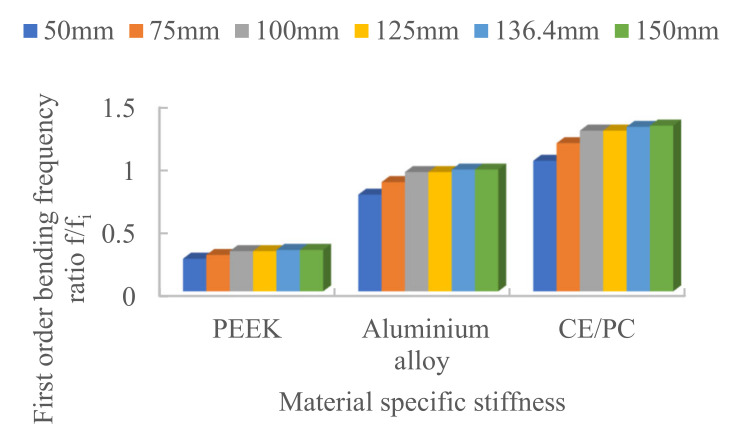
Effect of material specific stiffness on fundamental frequency.

**Figure 10 materials-15-06025-f010:**
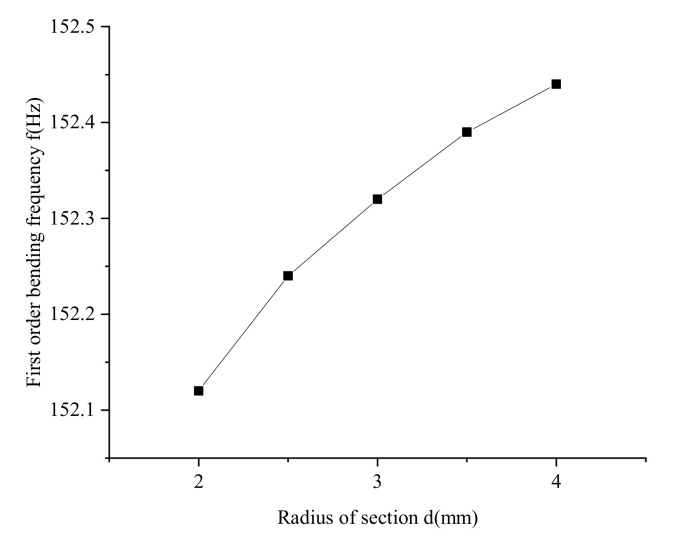
Influence of the cross-sectional diameter of the truss rod on the fundamental frequency.

**Figure 11 materials-15-06025-f011:**
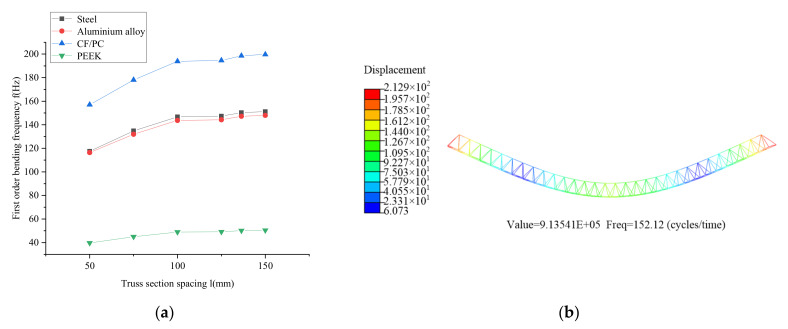
Effect of section spacing of the truss on the fundamental frequency: (**a**) Simulation comparison curve; (**b**) Simulation results.

**Figure 12 materials-15-06025-f012:**
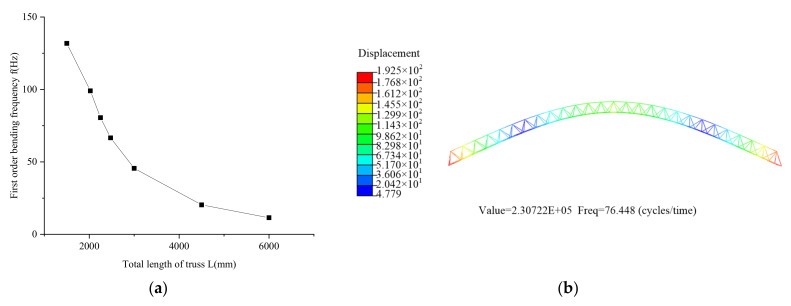
Effect of truss length on fundamental frequency: (**a**) Simulation comparison curve; (**b**) Simulation results.

**Figure 13 materials-15-06025-f013:**
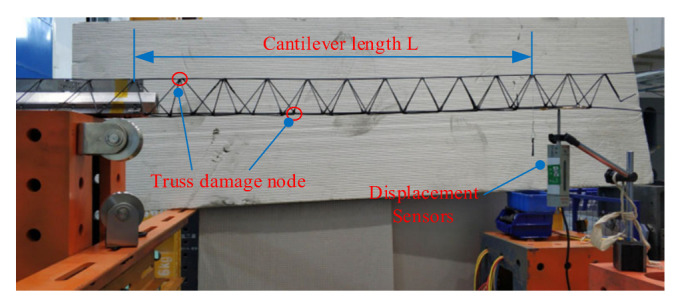
Static stiffness test of the truss.

**Figure 14 materials-15-06025-f014:**
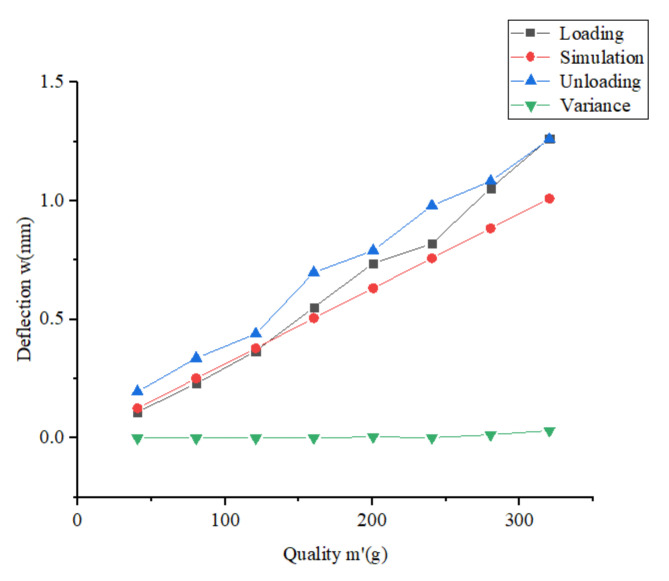
Static stiffness test curve of the truss.

**Figure 15 materials-15-06025-f015:**
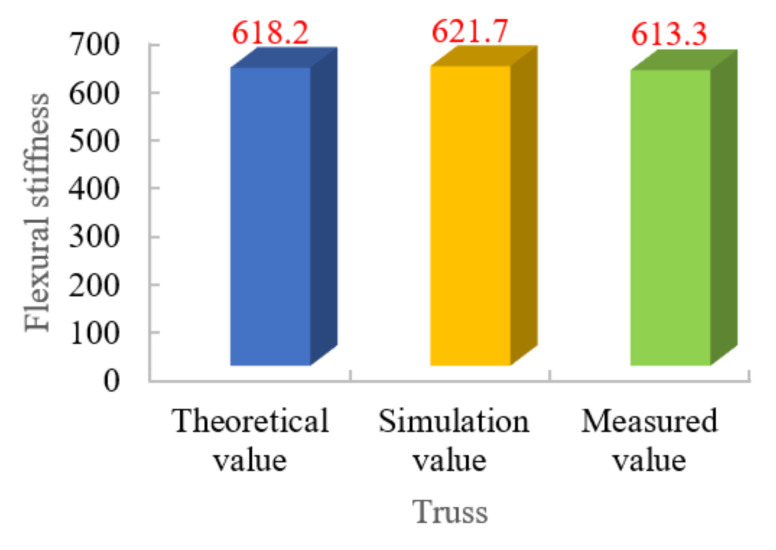
Comparison of flexural stiffness of trusses.

**Figure 16 materials-15-06025-f016:**
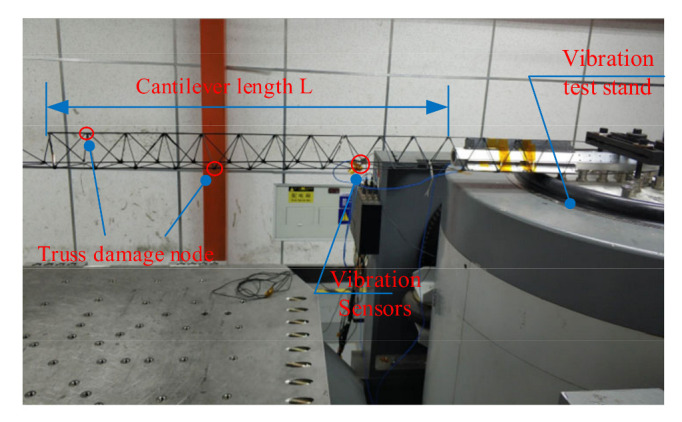
Truss vibration characteristics test.

**Table 1 materials-15-06025-t001:** Comparison table of common material properties for spacecraft manufacturing [22,23].

Materials	Glass State Transition Temperature or Melting Point (°C)	Thermal Conductivity (W/m·°C)	Low Temperature Environment Applicability	Manufacturing Process
PEEK	143	0.29	applicable	Hot Melt Molding
PC/CF	144	0.2	applicable	Hot Melt Molding
structural steel (45#)	1500	48.9	no applicable	Extrusion molding
aluminum alloy (7075)	650	228	no applicable	Extrusion molding

**Table 2 materials-15-06025-t002:** Comparison of test and simulation values of vibration characteristics of truss (Hz).

	First-Order	Second-Order	Third-Order	Fourth-Order
Test Value	72.4	74.5	200	200
Simulation value	76.44	76.45	203.41	203.49
Error	4.14%	2.62%	1.71%	1.75%

## Data Availability

Not applicable.

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
