# Peer review of "Design and Mechanical Characterisation of a Large Truss Structure for Continuous Manufacturing in Space"

_materials, 2022, doi:10.3390/ma15176025_

Round 1

Reviewer 1 Report (New Reviewer)

Please find attached the paper with remarks, suggestions and spelling corrections.

Author Response

Thank you for your advice.We have revised the paper based on the comments you made and finalized the final manuscript.Please see the attachment.

Reviewer 2 Report (New Reviewer)

This paper presents an analysis of the design and mechanical characteristics for a large truss structure

 The paper may be of interest to the scientific community through the topic addressed.

         The authors can take into account the following aspects:

- The introduction must be improved by taking into account other scientific papers in the field;

- of the 4 materials presented in Table 1, only two correspond to the criteria; I think that steel and aluminum should be eliminated, since the two materials do not behave well at low temperatures;

- The description of the ultrasonic welding process is very short and it is necessary to present the technological parameters of the process (vibration frequency, vibration amplitude, etc.);

- A clear specification of the boundary conditions that were imposed on finite element modeling in the case of simulation using the Abaqus software is required;

- It must be explained very clearly why there are large differences in values between the values determined theoretically, by simulation respectively measurement;

- The discussion part must be much developed in order to be able to highlight the novelty brought by the research presented in the paper in relation to other research in the field. In this form, the novelty of the research compared to previous research cannot be identified;

- Conclusions should be more concrete and future research directions should be presented.

Author Response

Thank you for your advice. We have revised the paper based on the comments you made and finalized the final manuscript.We put the paper and the response in the same PDF, with the response following the paper.

Reviewer 3 Report (New Reviewer)

A very good and valuable work has been done. But unfortunately, it is written in the form of a research report and not like a scientific article. In addition, the innovation of the work done is not bold compared to other researchers. On the other hand, there is a lack of scientific discussion in it and there is a need to interpret the obtained results. Therefore, I believe that this article has a good potential to be published in the journal and I encourage the authors to submit a new revision of the article to the journal. To improve the quality of the article, notice the following points:

1- Abstract should be rewritten and should include used methodology and important results. 

2- It is strongly recommended to edit final version of the manuscript by a Native English Editor. 

3- Literature review section should be improved and describe about lack of research done by others and show the advantage of the current research compared to others. 

4- The novelty of the present work should be written in the end paragraph of introduction. 

5- What is the reference for data reported in Table 1. 

6- Please add appropriate references for all equations. 

7-Simulation should be described in details such as element types, mesh sensitivity analysis, and ...

8- In figures 8 and 9, what is the meaning of 50, 75, 100, 125, 136.4, and150 and what is the unit. please correct it and it seems to be truss section spacing (mm). 

9- In figure 15, it is shown that the difference between simulation results and theoretical or measured data is very high and this error is not acceptable in industries. please describe your reason for this error? say about considering assumption in your simulation and improve your numerical research.

The points mentioned above are not all, but they are the most important ones that should be considered.

Author Response

Thank you for your advice. We have revised the paper based on the comments you made and finalized the final manuscript.We put the paper and the response in the same PDF, with the response following the paper.

Reviewer 4 Report (New Reviewer)

This work is of great interest. The work deals with the design of a one-dimensional truss structure and the mechanical properties that can be built continuously in space. Accordingly, simulations were performed and experimental verifications were also performed. The paper is informative. Therefore, I think, the paper as presented can be accepted for publication in Materials.

Author Response

Thank you for your advice. Thank you for reviewing Materials in your busy schedule and for your affirmation of our paper, which is also an affirmation of our work. With your affirmation, we have increased our confidence in this research and will continue our research in the direction of manufacturing in-space to achieve better results. We also wish you good health and success in the future. Thank you very much.

Round 2

Reviewer 2 Report (New Reviewer)

The authors revised their manuscript according to my suggestions. Thus the manuscript can be accepted for publication.

Author Response

Thank you for your advice. Thank you for reviewing Materials in your busy schedule and for your affirmation of our paper, which is also an affirmation of our work. With your affirmation, we have increased our confidence in this research and will continue our research in the direction of manufacturing in-space to achieve better results. We also wish you good health and success in the future. Thank you very much.

Reviewer 3 Report (New Reviewer)

It seems that the authors have best to provide revision manuscript based on the reviewer's comments and also they tried to response all comments one by one. However, I believe that the manuscript has a potential to publish in the current Journal but it is better to consider following points in order to improve the quality of the present form:

1- The reported results should be interpreted in more detail.

2- Finite element simulation should describe fully. 

3- The presentation should be more eloquent and scientifically expressed to get out of the report mode. 

4- It is better to extend literature review section to show more and comprehensive research done in this special issue and use more papers published recently. 

The presentation should be more eloquent and scientifically expressed to get out of the report mode. 

Author Response

Thank you for your advice. Please see the attachment.

This manuscript is a resubmission of an earlier submission. The following is a list of the peer review reports and author responses from that submission.

Round 1

Reviewer 1 Report

In this paper, Design and mechanical characterisation of a large truss structure for continuous manufacturing in-space, authors focuses on the truss configuration design and fundamental frequency analysis.

In the manuscript there are a few details, presented below, which require further attention by the authors.

 Comments and suggestions

 1. Lines 52, 53: Please use the section instead of "chapter"

2. Lines 117, 118: explain the acronyms PEEK, CF/PC

3. Leave space between values and units of measure (lines: 116, 155, etc.).

4. Line 116: Is the unit of measurement “m” or “mm”?

5. Figures 9 (b) and 10 (b) need to be improved because these (the data) are unclear

6. Please specify the software system used for the simulation

7. The results in Figures 13, 14 and not only must be discussed

Reviewer 2 Report

The authors made a methodical design error.

The authors, when designing a structure to work in space, should check in what temperature range it will work (negative and positive). Literature sources state that the lowest temperature can reach below minus 200 degrees Celcius. Structural materials and designs have completely different properties of working at such low temperatures. The authors should design structures and select mechanical properties for conditions in space. 

In the article, the authors should describe in detail the methodology of making the truss structure and the method of connecting the bars. With welded structures, there is a change in the structure and strength of the material at the connection points.

The authors should specify at the outset the requirements for design structures intended for use in space. Another element missing are references to standards and guidelines for the design of such structures.

Figures 9 and 10 are not legible. The authors should more extensively describe the restraint and loading of the truss in the FEM model. In Table 1, the authors miscounted the errors ( example- 203.49Hz-200Hz=3.49Hz- the authors give the unit %).

The authors study the frequency of vibration and its effect on the deflection arrow, and validated it using a static bending test. In my opinion, the authors should test the high-cycle durability of the structure.